# Construction of Watermelon Mutant Library Based on $^{60}$Co γ-ray Irradiation and EMS Treatment for Germplasm Innovation

Lijuan Yin [1], Yinjie Hou [1], Xiaoyao Chen [1], Xin Huang [1], Mengjiao Feng [1], Chunxia Wang [1], Zhongyuan Wang [1], Zhen Yue [1], Yong Zhang [1], Jianxiang Ma [1], Hao Li [1], Jianqiang Yang [1], Xian Zhang [1,2], Rong Yu [3,*] and Chunhua Wei [1,*]

[1] State Key Laboratory of Crop Stress Biology in Arid Areas, College of Horticulture, Northwest A&F University, Xianyang 712100, China; ylj980119@163.com (L.Y.); houyj1224@163.com (Y.H.); xiaoyaochen7186@163.com (X.C.); 18634350583@163.com (C.W.); yuezhen1225@126.com (Z.Y.); majianxiang@126.com (J.M.); yuanyilihao123@163.com (H.L.); yangjq1208@126.com (J.Y.)
[2] State Key Laboratory of Vegetable Germplasm Innovation, Tianjin 300384, China
[3] Institute of Horticulture, Ningxia Academy of Agriculture and Forestry Sciences, Yinchuan 750002, China
* Correspondence: yyrrhhyy@163.com (R.Y.); xjwend020405@nwafu.edu.cn (C.W.)

**Abstract:** Watermelon is a crucial horticultural crop worldwide but its genetic base has become extremely narrow owing to long-term cultivation. Induced mutagenesis can create a range of variations with distinctive agricultural characteristics. To broaden the genetic diversity of watermelon, we established a mutagenesis library containing over 4000 $M_1$ seeds from an inbred line 'M08', which was irradiated by 350 Gy of $^{60}$Co γ-rays for 3 h. The rates of germination, emergence, and survival of the $M_1$ seeds were reduced by 5.88%, 18.66%, and 41.96%, respectively. After phenotypic screening, 20 and 10 types of morphological changes were observed in the $M_1$ and $M_2$ generations, with approximately 10.57% and 14.17% mutation frequencies, respectively. Six mutants with desirable horticultural alterations were selected for additional presentation, including the leaf color mutant $C_1$-NO.1, the yellow peel mutant $C_1$-NO.2, the pericarp thickening mutant $C_1$-NO.3, the pericarp thinning mutant $C_1$-NO.4, the seedless mutant $C_1$-NO.5, and the $C_2$-No.1 mutant with normal female flowers and malformed male flowers. Moreover, the three mutants $M_1$-3, $M_2$-1, and $M_1$-5 were identified from our EMS-induced $M_2$ library, exhibiting the fusiform fruit, the dark green peel, and the yellow leaves, respectively. Compared to the wild type (WT), the photosynthetic pigments and parameters were negatively impacted in the yellow-leaf mutant $M_1$-5. For example, the total chlorophyll was 1.22 and 2.22 mg/g in the young and mature leaves of $M_1$-5, respectively, which were significantly lower than those in the WT (2.58 and 2.90 mg/g, respectively). Notably, some mutagenesis phenotypes could be stably inherited, including traits such as yellow leaf color, fusiform fruit shape, and thickening and thinning pericarp. Taken together, these results indicate that these two mutant libraries serve as essential resources to discover new phenotypic germplasms, thereby facilitating the genetic breeding and functional gene exploration in watermelon.

**Keywords:** watermelon; mutant library; morphological characterization; yellow leaf

## 1. Introduction

Watermelon (*Citrullus lanatus* L., 2n = 2x = 22) is a globally important cucurbit crop that is highly valuable and nutritious. For example, the abundant nutrients in watermelon, including vitamins, mineral salts, and amino acids (citrulline), provide numerous benefits for human health. Since it is one of the top five most consumed fresh fruits, watermelon had a global yield of 103.93 million tons in 2018, with 13.62 kg consumed per person [1]. China is the largest producer and consumer of watermelon in the world and it produced 63.0 million tons of watermelon cultivated on 1.51 million hm$^2$ in 2018 [1–3].

Watermelon has been reported to have originated in Africa and to have been cultivated for at least 4000 years [2,4]. Because of the long-term cultivation and selection for desirable fruit quality, modern watermelon cultivars currently have a narrow genetic base [3,5]. The genus *Citrullus* contains seven species [3,6]. To broaden the genetic diversity, three of these species (*C. colocynthis*, *C. amarus*, and *C. mucosospermus*) have been used to cross with cultivated watermelons in breeding programs [3,5,6]. However, the reproductive barriers and linkage drags limit the speed of genetic improvement in watermelon [3]. Therefore, the development of mutant libraries is considered to be a direct and effective approach to broaden the genetic base of watermelon and improve the breeding efficiency of economically important traits.

To date, several methods have been successfully utilized to construct plant mutant libraries, including physical ($^{60}$Co γ-rays and X-rays), chemical (ethyl methanesulfonate [EMS] and *N*-nitro-*N*-methylurea), and biological (T-DNA and transposons) approaches [7]. EMS has been widely adopted as a practical mutagen and is highly preferred owing to its high mutagenicity and ease of handling. For example, it has been used to create mutants in *Arabidopsis* [8], tomato (*Solanum lycopersicum* L.) [9,10], eggplant (*Solanum melongena* L.) [11], and strawberry (*Fragaria × ananassa*) [12]. Moreover, there are also numerous studies on EMS-induced mutant libraries in Cucurbitaceae crops. For example, EMS-induced mutant libraries in cucumber (*Cucumis sativus* L.) had been successfully constructed and some valuable germplasm resources were subsequently identified, including those related to flower size, leaf color, and fruit shape and peel [7,13]. Importantly, several regulatory genes for agronomic traits were cloned using EMS-induced mutant lines, including the gene for yellow leaf *CscpFtsY* [14], unusual flower and tendril *CsUFO* [15], and curly leaf *CsPHB* [16]. In melon (*Cucumis* sp.), the *CRTISO* gene responsible for the accumulation of carotenoids was also cloned from an EMS-induced mutant '*yofl*' with yellow-orange flesh [17]. In watermelon (*Citrullus lanatus* L.), a photosensitive flesh line '*psf*' was identified from the EMS-induced mutant library; a G-A transversion leads to a premature stop codon in the causal gene *ClZISO*, which results in morphological changes [18]. A pollen-EMS mutagenesis method was used to construct a library that contained 200,000 $M_1$ seeds and two genes responsible for fruit shape and male sterility were identified, respectively [3]. Similarly, $^{60}$Co γ-rays have also been used to construct mutant libraries in plants. For instance, 10 accessions of tetraploid wheat were irradiated with $^{60}$Co γ-rays, which resulted in several novel mutation resources for wheat breeding [19]. The efficiency of different doses of $^{60}$Co γ-rays was extensively investigated in tomato and potato (*Solanum tuberosum* L.) [10,20]. Similarly, $^{60}$Co γ-rays were also utilized to induce mutations in cucurbit crops, such as bitter gourd (*Momordica charantia* L.) [21], squash (*Cucurbita* spp.), and pumpkin (*Cucurbita moschata* Duchesne) [22]. Although various doses of $^{60}$Co γ-rays have been used to determine the optimal dose to construct the watermelon library [23,24], a large-scale mutant library has not yet been created.

In this study, a watermelon mutant library that contained 4000 $M_1$ seeds was successfully constructed using $^{60}$Co γ-ray irradiation. After undergoing morphological characterization, a serious of phenotypic mutants were identified, with 20 and 10 types of morphological changes observed in the $M_1$ and $M_2$ generations, respectively. Taken together with the mutant library established by our group through EMS treatment [25,26], nine representative mutants were characterized in detail. It is notable that some mutagenesis phenotypes could be stably inherited, including traits such as yellow leaves, fusiform fruit shape, and thickening and thinning pericarps. In summary, these two watermelon mutant libraries including the nine well-represented mutants can provide promising germplasm resources to breed new varieties of watermelon and explore novel gene functions.

## 2. Materials and Methods

### 2.1. Plant Materials

The watermelon germplasm 'M08' used in this study was provided by the Watermelon and Melon Research Group at Northwest A&F University (Yangling, China) [27]. To con-

struct a mutagenesis library, dry seeds were irradiated with 350 Gy of $^{60}$Co γ-rays for 3 h at the Hefei Institute of Physical Sciences, Chinese Academy of Sciences, Hefei, China.

### 2.2. Construction of the Mutant Library

To obtain a large-scale $M_1$ library, approximately 2000 irradiated seeds were independently germinated in the spring of 2021 and 2022, factoring in the constraints of available experimental field space and human resources. After the germination rate had been calculated (number of germinated seeds/total number of seeds × 100%), the seeds were sown in plastic trays with 50 holes filled with commercial peat-based compost. In addition, the emergence rate and seedling rate were recorded according to the relevant index calculation formulae previously described [28]. The seedlings were then transferred to farms at Northwest A&F University for morphological screening and self-pollination to produce the $M_2$ generation. Among the $M_2$ families, we randomly selected 17 independently self-crossed lines and grew 15 plants for each line. Morphological observations of the $M_2$ plants were also performed in the greenhouses of Northwest A&F University.

As described in our previous studies [25,26], we established an EMS mutant library using the watermelon line 'M08' as the material. To validate the morphological variants of three valuable mutants ($M_1$-3, $M_2$-1, and $M_1$-5), their $M_2$ offspring were subsequently planted at the farms of Northwest A&F University (Yangling, China).

### 2.3. Phenotypic Characterization of the Mutants

The morphological changes were identified visually during the whole duration of the developmental stages. For the cotyledon phenotype, the color, size, and number were primarily inspected when the plantlets were grown in trays. The phenotypic alterations of other agricultural traits, such as the leaf morphologies, plant architecture, floral organs, and fruit changes, were recorded after the seedlings had been transferred to the field. A unique ID was assigned to each mutant to correlate with their phenotypic characteristics. Normal plants of germplasm 'M08' were used as the control to identify mutants.

### 2.4. Measurement of the Physiological Indices of Mutant $M_1$-5

At the stem elongation stage, we selected the young and mature leaves from the mutant $M_1$-5 and wild type (WT) to analyze the contents of pigment and photosynthetic characters. Fresh leaves were cut into small pieces and 0.1 g samples were placed into 10 mL 95% ethanol at room temperature for 24 h. After centrifugation at $3500\times g$ for 10 min, the contents of chlorophyll a (Chl a), chlorophyll b (Chl b), total chlorophyll (Chl a + b), and carotenoids (Caro) were measured colorimetrically at 665 nm, 649 nm, and 470 nm using a spectrophotometer, as previously described [29,30]. Three technical replicates were performed for each sample. A volume of 95% ethanol was used as the blank control. Moreover, photosynthesis-related indices, including the net photosynthetic rate (Pn), stomatal conductance (Gs), intercellular carbon dioxide concentration (Ci), and transpiration rate (Tr), were measured using an LI-6800 portable photosynthetic system (LICOR, Lincoln, NE, USA). The conditions were a temperature of 25 ± 2 °C, a $CO_2$ concentration of 380 μmol mol$^{-1}$, and a photosynthetic photon flux density (PPFD) of 500 μmol m$^{-2}$ s$^{-1}$, as previously described [1]. The amounts of pigments were calculated using the following equations:

$$Ca = 13.95 \times A665 - 6.88 \times A649$$

$$Cb = 13.95 \times A649 - 6.88 \times A655$$

$$Chl\ a = (Ca \times V)/(W \times 1000)$$

$$Chl\ b = (Cb \times V)/(W \times 1000)$$

$$Caro = (1000 \times A_{470} - 2.05 \times Ca - 114.8 \times Cb)/245$$

$A_{665}$, $A_{649}$, and $A_{470}$ were the absorbance values at wavelengths 665 nm, 649 nm, and 470 nm, respectively. V and W represented the volume of extracted liquid and sample weight, respectively.

### 2.5. Statistical Analysis

All of the data were presented as the mean $\pm$ SD for at least three independent replicates. The significance was analyzed using a one-way analysis of variance (ANOVA) with a Duncan's test ($p < 0.05$) via SPSS 25.0 (IBM, Inc., Armonk, NY, USA).

## 3. Results

### 3.1. Negative Effect of $^{60}$Co-$\gamma$ Radiation on Seed Viability

After the seeds had been irradiated with $^{60}$Co $\gamma$-rays, their viability was evaluated to assess the effect of a dosage of 350 Gy radiation. Compared to the normal seeds, the germination rate, the emergence rate, and the seedling rate were obviously reduced by 5.88%, 18.66%, and 41.96%, respectively (Table 1). These effects suggested that the $^{60}$Co radiation had a negative impact on seed vigor. In addition, during the germination process, we observed that the normal seeds began germinating after approximately 24 h in the incubator; whereas, the seeds exposed to $^{60}$Co-$\gamma$ radiation generally started germinating after 36 h, indicating that the $^{60}$Co-$\gamma$ radiation delayed and inhibited seed germination.

**Table 1.** Effects of $^{60}$Co-$\gamma$ radiation on the germination rate, emergence rate, and seedling rate of watermelon.

| Material | Germination Rate (%) | Emergence Rate (%) | Seedling Rate (%) |
|---|---|---|---|
| M08 | 95.45% | 90.91% | 88.18% |
| $M_1$ | 89.84% | 73.95% | 51.18% |

### 3.2. Observation and Analysis of the $M_1$ and $M_2$ Generations

To construct a large library of $M_1$ mutants, approximately 2000 seeds were sown independently in 2021 and 2022 and 952 and 903 individuals were obtained respectively. Among the $M_1$ population (Table 2), 196 mutant plants with phenotypic variations were identified and they exhibited 20 different types of mutations based on the characteristics of plant architecture, leaves, and flowers. The overall mutation frequency was approximately 10.57% while those for the leaves, stems, flowers, and fruits were 4.74%, 0.11%, 4.37%, and 1.13%, respectively. Among the mutants of leaf organs, several plants with crumpled leaf edges and yellowed leaves were discovered (Figure 1). However, in the mutants of flower organs, variations were observed in the number of petals and the presence of bisexual flowers.

**Table 2.** Summary of mutant types in the $^{60}$Co $\gamma$-ray-induced library in the $M_1$ generation.

| Traits | Mutant Description | Number of Plants | Mutation Frequency (%) |
|---|---|---|---|
| Leaf | Yellow leaf | 7 | 0.38 |
| | Pale gray leaf | 4 | 0.22 |
| | Chimera color leaf | 6 | 0.32 |
| | Deformed leaf | 66 | 3.56 |
| | Large leaf | 2 | 0.11 |
| | Small leaf | 3 | 0.16 |
| Stem | Shorter internode | 2 | 0.11 |
| Flower | Bisexual flower | 40 | 2.16 |
| | Variation in petal numbers in male flowers | 15 | 0.81 |
| | Variation in petal numbers in female flowers | 12 | 0.65 |
| | Petal deformity | 10 | 0.54 |
| | Clustered male flowers | 2 | 0.11 |
| | Male sterility | 2 | 0.11 |

**Table 2.** *Cont.*

| Traits | Mutant Description | Number of Plants | Mutation Frequency (%) |
|---|---|---|---|
| Fruit | Stripe pattern variation | 6 | 0.32 |
| | Fruit color variation | 7 | 0.38 |
| | Fruit shape variation | 4 | 0.22 |
| | Flesh color variation | 1 | 0.05 |
| | Pericarp thickness variation | 2 | 0.11 |
| | Seedless | 1 | 0.05 |
| Others | Slow-growing | 4 | 0.22 |

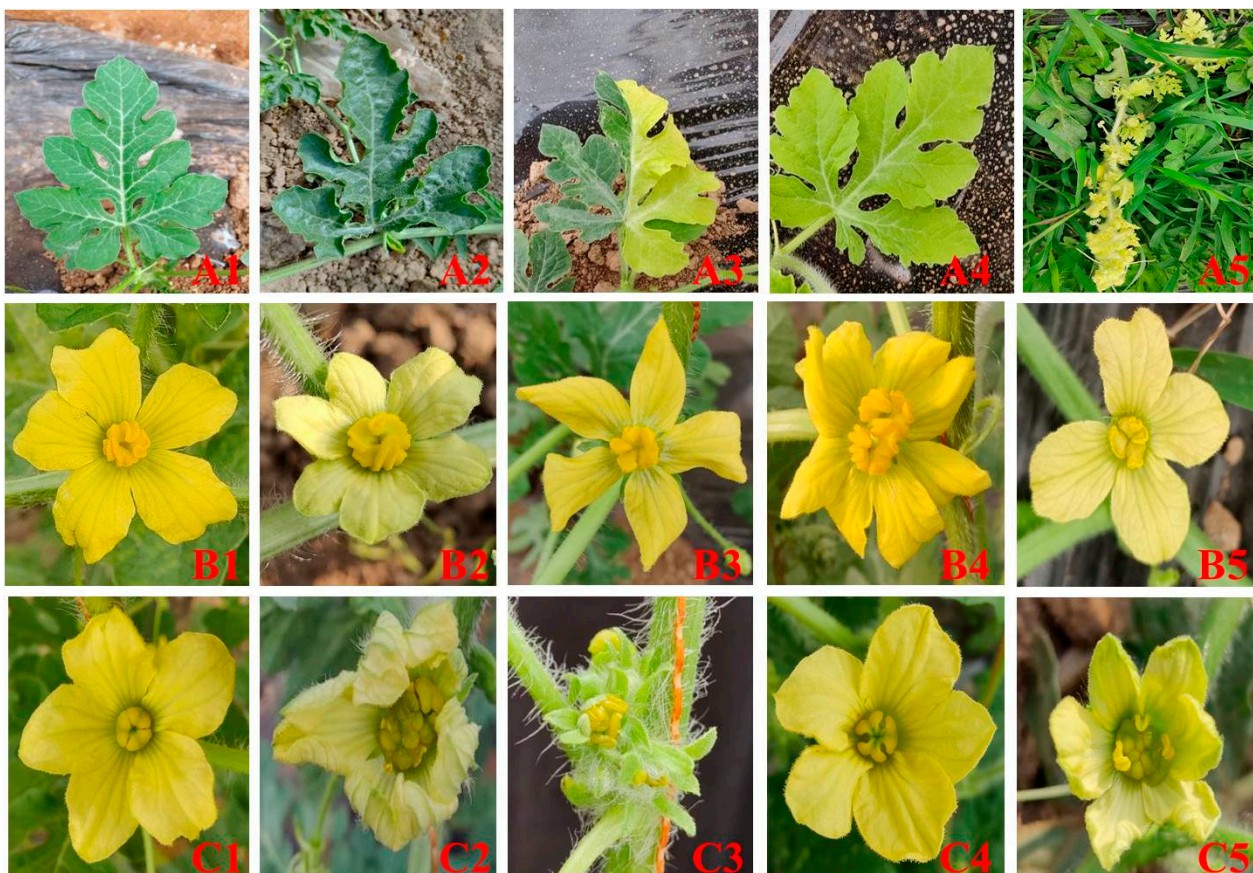

**Figure 1.** Representative mutant phenotypes in the M$_1$ generation. (**A1**): Leaf (WT); (**A2**): Crumpled leaf edges; (**A3**): Chimera color leaf; (**A4**): Yellow leaf; (**A5**): Yellow leaf. (**B1**): Male flower (WT); (**B2**): Increased number of pale-yellow petals; (**B3**): Narrow petal shape; (**B4**): Deformed petals and stamens; (**B5**): Decreased number of pale-yellow petals. (**C1**): Female flower (WT); (**C2**): Deformed petals and stigmas; (**C3**): Clustered flowers; (**C4**): Deformed stigmas; (**C5**): Bisexual flower.

We selected 17 self-crossed lines of the M$_2$ population. Each consisted of 15 plants. A total of 247 plants ultimately survived. Among them, 10 types of mutations involving 35 individuals were identified, with a mutation frequency of 14.17% (Table 3). Compared to the normal plants, these mutations involved variant cotyledons, leaf margins, and pericarps (Figure 2). For example, we observed that one line had completely yellow cotyledon individuals during the seedling stage (Figure 2B) and the ratio of yellow plants to normal plants was approximately 1:4 (4 yellow cotyledon and 16 normal seedlings). However, the yellow plants grew significantly weaker and they gradually wilted and eventually died, possibly owing to a decrease in their photosynthetic capacity.

**Table 3.** Summary of mutant types in the $^{60}$Co γ-ray-induced library in the M$_2$ generation.

| Traits | Mutant Description | Number of Plants | Mutation Frequency (%) |
|---|---|---|---|
| Cotyledon | Cotyledon deformity | 1 | 0.4 |
| | Yellow cotyledon | 4 | 1.62 |
| | Yellow-green cotyledon | 1 | 0.4 |
| Leaf | Deformed leaf | 9 | 3.64 |
| | Deep-lobed leaf | 2 | 0.81 |
| Stem | Shorter internode | 1 | 0.4 |
| Flower | Bisexual flower | 4 | 1.62 |
| | Delayed flowering | 1 | 0.4 |
| | Male sterility | 1 | 0.4 |
| Fruit | Variation in pericarp thickness | 11 | 4.45 |

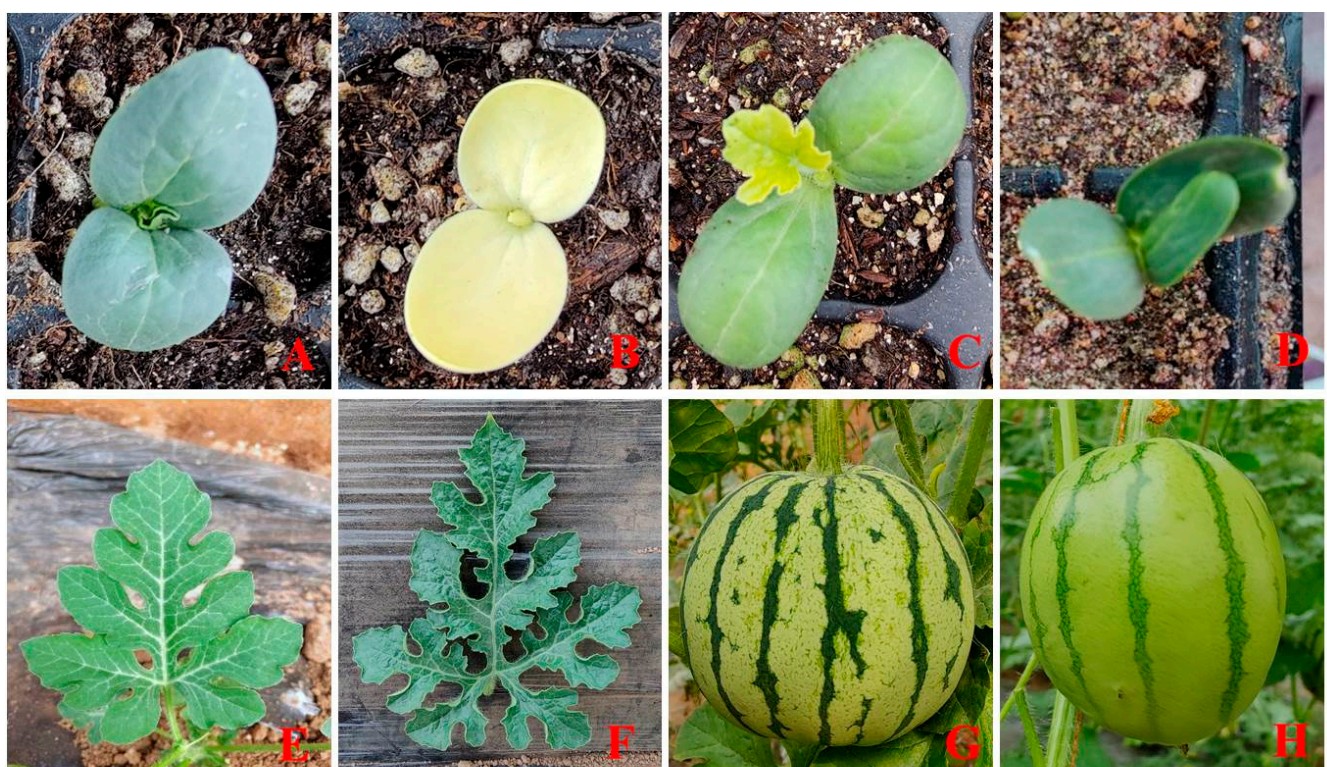

**Figure 2.** Representative mutant phenotypes in the M$_2$ generation. (**A**): Cotyledon (WT); (**B**): Yellow cotyledon; (**C**): Yellow leaf; (**D**): Deformed cotyledon; (**E**): Leaf (WT); (**F**): Crumpled leaf edges; (**G**): Young fruit (WT); (**H**): Light stripes; WT: wild type.

*3.3. Morphological Screening of the Mutations Identified by $^{60}$Co γ-ray Irradiation and EMS Treatments*

As described above, hundreds of mutants were primarily identified from the M$_1$ and M$_2$ generations. They exhibited various changes in phenotype. Considering the fruits as an important trait of watermelon, we primarily narrowed the search to mutants related to this character. Among all of the morphological mutations observed following $^{60}$Co-γ irradiation, we selected five from the M$_1$ population and one from the M$_2$ population for details. They were designated C$_1$-No.1, C$_1$-No.2, C$_1$-No.3, C$_1$-No.4, C$_1$-No.5, and C$_2$-No1. The C$_1$-No.1 plant was a mutant with yellow leaves during the growth period (Figure 3A,B). It harbored abnormal male flowers that made it difficult for self-pollination to produce the fruits. Compared to the WT, the pericarp of C$_1$-No.2 was light green at the early stage of fruit development and it gradually became a milky white at the mature period (Figure 3C–E). The normal rind thickness of germplasm M08 was approximately

1.1 cm. In contrast, the rind thickness of the mutants $C_1$-No.3 and $C_1$-No.4 were 2.4 cm and 0.4 cm, respectively, which represented an increase of 118.18% and a decrease of 63.64%, respectively (Figure 3F–H). As a seedless mutant, $C_1$-No.5 contained only one seed and had a lighter pink flesh compared to the WT (Figure 3F,I). More importantly, the mutant traits of $C_1$-No.3 and $C_1$-No.4 had been proven to be stably inherited through their self-crossed offspring (Figure S1A,B). In the $M_2$ generation, a new male-sterile mutant, designated $C_2$-No.1, was discovered with normal female flowers and malformed male flowers (Figure S1C).

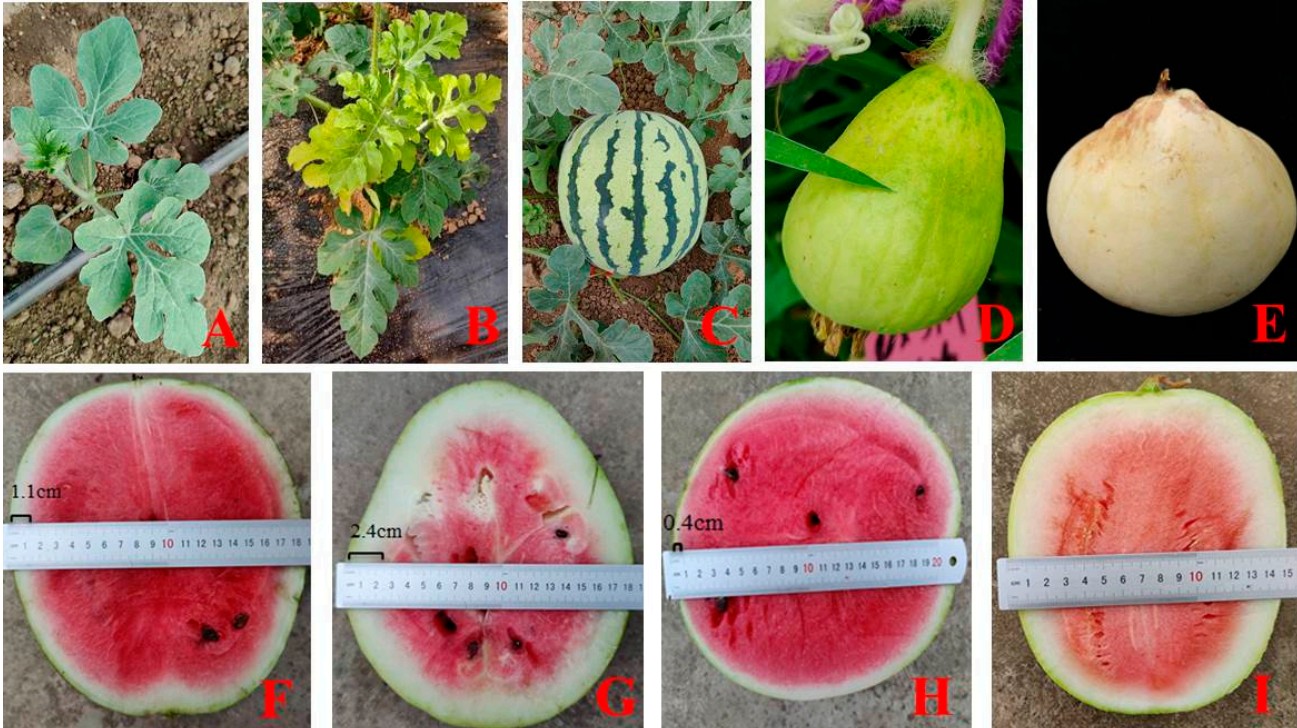

**Figure 3.** Five representative mutants from a $^{60}$Co γ-ray-induced library. (**A**): Normal seedling (WT); (**B**): $C_1$-NO.1; (**C**): Mature fruit (WT); (**D**,**E**): $C_1$-NO.2, young and mature stages, respectively; (**F**): Mature fruit (WT); (**G**): $C_1$-NO.3; (**H**): $C_1$-NO.4; (**I**): $C_1$-NO.5; WT: wild type.

We used watermelon germplasm 'M08' to construct an EMS mutant library in our previous studies [23,24] and identified some valuable mutants in the $M_1$ generation. In this study, we presented three of them, which were designated $M_1$-3, $M_2$-1, and $M_1$-5. The mutant $M_1$-3, which was characterized by fusiform fruit morphology in the $M_1$ generation, had been proven to be stably inherited in the $M_2$ offspring (Figure 4A,B). Compared to the WT, the $M_2$-1 plant fruits were dark green with clear stripes (Figure 4A,C). In addition, as a leaf color mutant, the new young leaves of $M_1$-5 were yellow while its mature leaves gradually turned green at the stem elongation stage. However, the mature leaves were still lighter than those of the WT (Figure 4D,E).

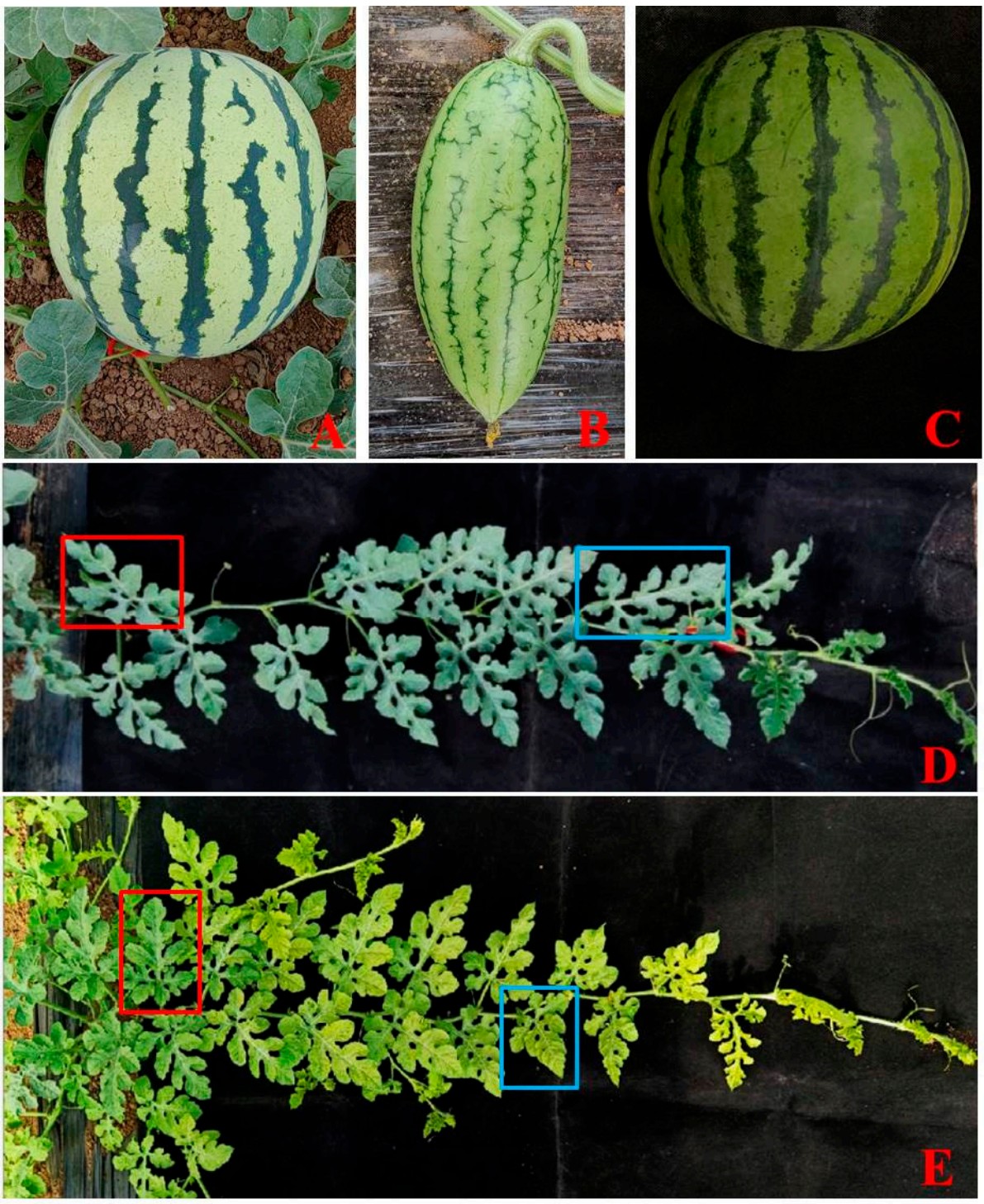

**Figure 4.** Three representative mutants from the EMS-induced library. (**A**): Mature fruit (WT); (**B**): $M_1$-3; (**C**): $M_2$-1; (**D**): Seedlings (WT); (**E**): $M_1$-5. Blue boxes: young leaves; Red boxes: mature leaves. EMS: ethyl methanesulfonate; WT: wild type.

*3.4. Statistical Analyses of the Contents of Photosynthetic Pigments and Photosynthetic Parameters in Mutant $M_1$-5*

Leaf color may not only affect watermelon yield but can also serve as a morphological marker to assess the authenticity of hybrids. Hence, we further investigated the photosynthetic pigment content and photosynthetic parameters of $M_1$-5 at the stem elongation stage. As expected, the content of carotenoids and total chlorophyll, including chlorophyll

a and b, were consistently lower in the young leaves compared to the mature leaves in both the $M_1$-5 and WT plants (Table 4). In addition, the ratios of Chl a/b and Caro/Chl in the new leaves of $M_1$-5 were significantly higher than those in the mature leaves. However, the ratios did not change much in the WT, which was consistent with the changes in leaf color (Table 4 and Figure 4D,E). Furthermore, we investigated the differences in Pn, Gs, Tr, and Ci between the WT and $M_1$-5. Compared with that of the WT, the Pn of young and mature $M_1$-5 leaves decreased by 32.58% and 24.52%, respectively, which was similar to the trend of Gs (Figure 5A,B). The Ci indicated that the concentration of carbon dioxide ($CO_2$) in $M_1$-5 was significantly higher than that in the WT, suggesting that the mutant utilized a lower rate of $CO_2$ (Figure 5C). In addition, the Tr of mature leaves in the WT was obviously stronger than that in the other leaves (Figure 5D). Taken together, these results indicated that the photosynthetic pigments and parameters were negatively impacted in the $M_1$-5 mutant.

**Table 4.** Contents of photosynthetic pigments in the young and mature leaves of the mutant M1-5 and the WT.

| Leaf | Material | Chl a (mg/g) | Chl b (mg/g) | Caro (mg/g) | Chl (mg/g) | Ratio of Chl a/b | Ratio of Caro/Chl |
|---|---|---|---|---|---|---|---|
| Young leaves | WT | 1.90 ± 0.01 b | 0.67 ± 0.03 b | 0.64 ± 0.01 b | 2.58 ± 0.02 b | 2.82 ± 0.13 c | 0.25 ± 0.01 c |
|  | M1-5 | 1.00 ± 0.02 d | 0.22 ± 0.02 d | 0.59 ± 0.02 c | 1.22 ± 0.02 d | 4.52 ± 0.38 a | 0.48 ± 0.02 a |
| Mature leaves | WT | 2.15 ± 0.02 a | 0.75 ± 0.01 a | 0.74 ± 0.03 a | 2.90 ± 0.02 a | 2.86 ± 0.01 c | 0.25 ± 0.08 c |
|  | M1-5 | 1.73 ± 0.02 c | 0.48 ± 0.02 c | 0.67 ± 0.03 b | 2.22 ± 0.01 c | 3.59 ± 0.16 b | 0.30 ± 0.01 b |

Note: Caro: carotenoids; Chl: chlorophyll; Chl a: chlorophyll a; Chl b: chlorophyll b; WT: wild type. The different letters indicate significance according to the Duncan's test ($p < 0.05$).

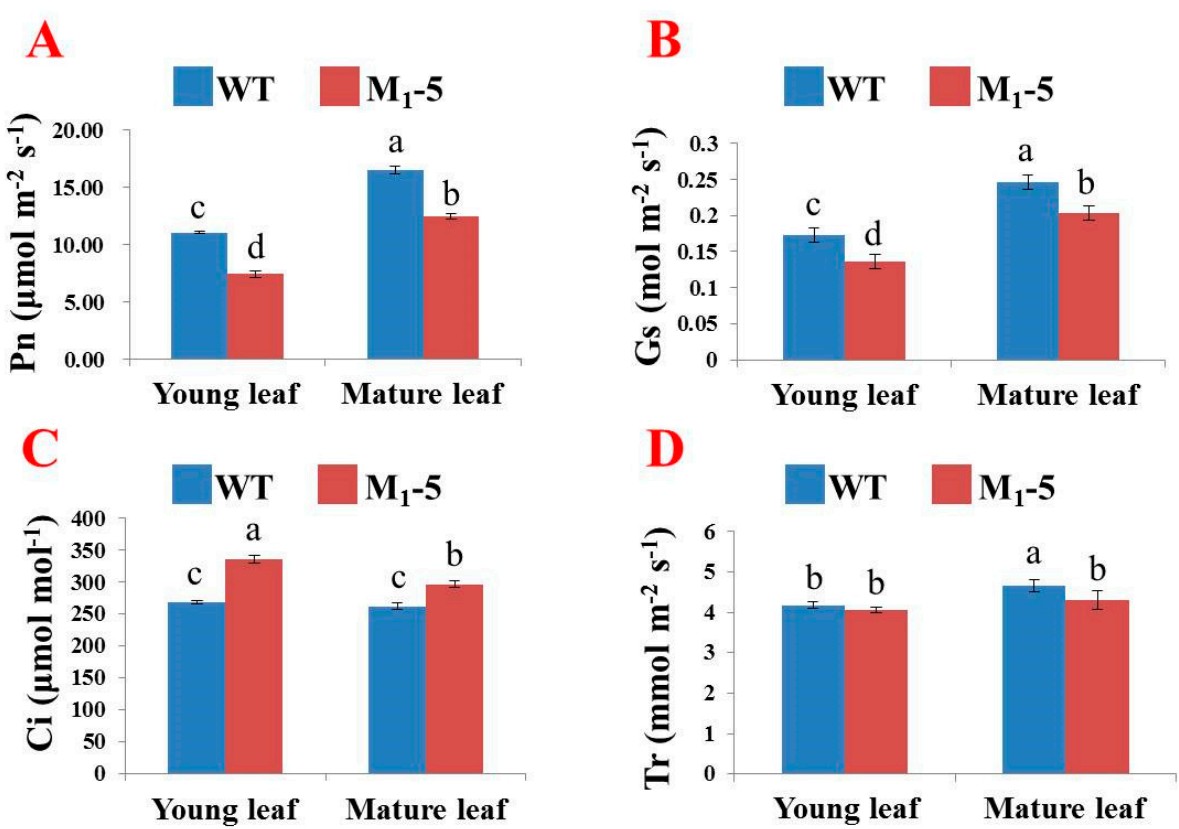

**Figure 5.** Photosynthetic parameters of the WT and yellow-leaf mutant $M_1$-5, including the Pn (**A**), Gs (**B**), Ci (**C**), and Tr (**D**). The data are represented as the mean ± SD. The different letters indicate significance according to the Duncan's test ($p < 0.05$). Pn: net photosynthetic rate, Gs: stomatal conductance, Ci: intercellular $CO_2$ concentration, Tr: transpiration rate; WT: wild type.

## 4. Discussion

Watermelon has been cultivated for at least 4000 years, which has led to a very low degree of genetic diversity [2,4]. Thus, the construction of a mutant library will provide potential favorable germplasm to breed new varieties and resources to explore gene functions. Although EMS has been widely used as a stable and effective mutagen, the amount of the mutagen that is lethal to one-half of the experimental population ($LD_{50}$) and the efficiency of mutagenesis vary among plant species. For example, the treatment of 1% EMS for 24 h was much closer to the $LD_{50}$ for the cucumber cultivar 'Shannong No.5', which resulted in an 18.3% frequency of mutation [13]. However, the 1.5% EMS treatment for 12 h was optimal for the cucumber inbred line '406' based on the $LD_{50}$ [7]. For the watermelon line '302', the mutants were induced by 1.5% EMS for 10 h [18] while 1.2% EMS for 12 h was the best treatment to construct a mutant library of the watermelon germplasm 'Shihong' [31]. In our previous research [25,26], we established a watermelon mutant library using the inbred line 'M08' treated with 1.5% EMS for 12 h and obtained a series of mutagenesis lines in the $M_1$ progenies with a total mutation frequency of 20.56%, which is much higher than that of the pollen-EMS mutation library of 15.38% [3].

Similarly, $^{60}$Co γ-rays have also been widely utilized in plant breeding to induce the desired phenotypic mutagenesis and various species respond differentially to a suitable dose of radiation. For instance, a dose of radiation that ranged from 200 to 250 Gy displayed much higher mutagenic efficiency in bitter gourd [21] compared to the safely recommended doses between 200 and 250 Gy in the winter squash and pumpkin lines [22]. In watermelon, dry seeds of the variety 'Bojura' were treated with $^{60}$Co γ-rays at 80, 100, 200, 250, 350, and 450 Gy, respectively, revealing that the survival rate (66.20% and 62.07%) of the higher doses (350 and 450 Gy) was close to the $LD_{50}$ [23]. Similarly, the germination rate of the watermelon genotype 'Yalıncak' was 51.11% after exposure to 350 Gy of $^{60}$Co γ-rays [24]. Nevertheless, to our knowledge, a large-scale mutant library has not been generated using the $^{60}$Co γ-rays irrigation method to date. In this previous study, 4000 seeds of the watermelon inbred line 'M08' were used to construct an irradiation mutant library. The rate of seed germination was approximately 89.84% (Table 1), which was much higher than those of the genotypes 'Yalıncak' and 'Bojura' [23,24], despite the fact that a similar dosage of $^{60}$Co γ-rays (350 Gy) was applied. Since research to determine the optimal mutation dose in watermelon is currently limited, studies in other species of vegetables suggest that higher dosages of irradiation have an adverse effect on plant growth [24]. In this study, the rate of seeds from the irradiated line 'M08' was only 51.18% (Table 1), which was slightly lower than that of the genotype 'Bojura' at dosages of 350 Gy (66.20%) and 450 Gy (62.07%) [23]. Taken together, these data indicate that the suitable dosage varies among genotypes and higher levels of $^{60}$Co γ-ray irradiation have a negative effect on both seed germination and plant growth.

In general, most of the mutants induced by EMS or $^{60}$Co γ-rays may be harmful to plants or unfavorable from a breeding perspective. For example, we found a yellow-cotyledon mutagenic line in the $M_2$ families (Figure 2B) and four yellowing seedlings were observed among a total of 20 plants, suggesting that this phenotypic change in morphology is controlled by a recessive gene. However, the mutagenic seedlings gradually withered and eventually died, possibly owing to their decrease in photosynthetic capacity. Undoubtedly, some mutants indeed exhibit desirable horticultural traits that can be used for the further breeding of new varieties or the exploration of gene functions. For example, fruit shape is one of the major objectives in watermelon breeding, which can influence consumer preference and fruit transportation. In the pollen-EMS mutant library, a mutant with elongated fruit was identified to be incompletely dominant compared to the spherical fruit [3]. In this study, we also discovered a fruit mutant, $M_1$-3, which exhibits a stably inherited fusiform fruit shape (Figure 4A,B). This phenotype is different from the previously published elongated fruit shape [3], indicating that mutant $M_1$-3 is a new type of germplasm. However, further genetic analysis and functional gene mining of this morphological trait are needed to investigate this hypothesis. Male sterility has been

recognized as a useful trait to utilize hybrid vigor and produce hybrid seeds in watermelon breeding. To date, only one dominant and two recessive male sterile genes have been cloned from watermelon mutants [3,32,33]. In this study, we also reported a male-sterile mutant, $C_2$-No.1, that produces malformed male flowers (Figure S1C), which has potential applications in the utilization of watermelon hybrid vigor. The colorful flesh of watermelon can attract consumers and benefit human health. Using the EMS-induced mutant '*psf*', the *ClZISO* gene was predicted to be responsible for the photosensitive flesh, thus providing the theoretical basis for color breeding in watermelon [18]. In this study, a different mutant, $C_1$-No.5, was characterized with a lighter pink flesh and seedless phenotype (Figure 3F,I), implying its significant value in the exploration of gene function and breeding of new varieties. Additionally, other mutants with stably inherited phenotypes, such as the rind thickness mutants $C_1$-No.3 and $C_1$-No.4 (Figure 3F–H), are also new valuable germplasms to develop new varieties of watermelon with favorable characteristics.

The morphological changes in leaves are a typical observation in mutant libraries. In both the $M_1$ and $M_2$ generations, the leaf-type variant mutants were the majority in our study (Tables 2 and 3), which is consistent with the findings in other species, such as cucumber [7], strawberry [12], and eggplant [11]. Among these leaf-type mutants, changes in color were the most common mutants. For example, leaf mutants accounted for the highest proportion in the cucumber EMS-induced library and the changes in color varied from light green to pale yellow and dark green [7]. Considering that leaves serve as crucial organs to produce photosynthates for plant growth, leaf-color-type mutants are recognized as vital types of germplasm to study chloroplast and photomorphogenesis. In watermelon, a chlorophyll-deficient mutant '*Yl2*' with yellow leaves was identified from an EMS-induced library; an observation of its anatomy revealed that the yellow leaves had fewer chloroplasts and thylakoids [34]. In this study, a mutant designated $M_1$-5 was obtained; its young leaves were yellow but the mature leaves gradually turned green at the stem elongation stage (Figure 4D,E). These changes are similar to the phenotype of a natural virescent-leaf-color mutant '63' [35]. According to the data of photosynthetic pigments and parameters between the $M_1$-5 and WT (Table 4 and Figure 5), we deduced that the development of the chloroplast and efficiency of the photosynthetic performance in the former were possibly inhibited, which is consistent with the findings in mutants *Yl2* and '63' [34,35].

## 5. Conclusions

We established a large-scale watermelon mutant library using a treatment of irrigation with $^{60}$Co γ-rays. Hundreds of mutants that exhibited various phenotypic changes were identified from the $M_1$ and $M_2$ generations. In detail, six mutants related to fruit ($C_1$-NO.2, $C_1$-NO.3, $C_1$-NO.4, $C_1$-NO.5, $M_1$-3, and $M_2$-1), along with one flower ($C_2$-No.1) and two leaf color ($C_1$-NO.1 and $M_1$-5) mutants, were additional mutants. The analysis of photosynthetic pigment and parameters between the WT and yellow-leaf mutant, $M_1$-5, at the stem elongation stage indicated that the development of chloroplasts and the efficiency of photosynthetic performance were possibly inhibited in this mutant. Notably, the yellow leaf, as well as the fusiform fruit shape and thickening and thinning pericarp, can be stably inherited, which can be used in the genetics of watermelon breeding and the exploration of functional genes. Moreover, the large-scale watermelon mutant library could provide more valuable germplasm resources for further research.

**Supplementary Materials:** The following supporting information can be downloaded at: https://www.mdpi.com/article/10.3390/horticulturae9101133/s1, Figure S1, Phenotypes of $C_1$-NO.3 (A), $C_1$-NO.4 (B), and $C_2$-No.1 (C) in the $M_2$ generation.

**Author Contributions:** Conceptualization, L.Y.; methodology, R.Y. and C.W. (Chunhua Wei); software, L.Y. and Y.H.; validation, X.C., X.H., M.F., C.W. (Chunxia Wang), Z.W. and Z.Y.; formal analysis, L.Y. and C.W. (Chunxia Wang); investigation, L.Y., Y.H., M.F., X.C. and Z.W.; resources, Y.Z., J.M. and J.Y.; writing—original draft preparation, L.Y. and C.W. (Chunhua Wang); writing—review and editing, X.Z., R.Y. and C.W. (Chunhua Wei); visualization, X.Z.; supervision, H.L., R.Y. and C.W. (Chunhua Wei); project administration, R.Y. and C.W. (Chunhua Wei); funding acquisition, R.Y. and C.W. (Chunhua Wei). All authors have read and agreed to the published version of the manuscript.

**Funding:** This work was supported by the High-quality Development and Ecological Protection Science and Technology Innovation Project of Ningxia Academy of Agriculture and Forestry Sciences (NGSB-2021-7), the Seed Innovation Project of Northwest A&F University (2452022116), the National Natural Science Foundation of Shaanxi Province, China [No. 2023-JC-YB-199], and the Key Research and Development Project of Yangling Seed Industry Innovation Center (Ylzy-sc-01).

**Data Availability Statement:** All relevant data can be found within this manuscript and the supporting information files.

**Conflicts of Interest:** The authors declare no conflict of interest.

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
