# Peer review of "Construction of Watermelon Mutant Library Based on 60Co γ-ray Irradiation and EMS Treatment for Germplasm Innovation"

_horticulturae, doi:10.3390/horticulturae9101133_

Round 1

Reviewer 1 Report

-          This paper correspond for scope of journal. 

-          The title corresponds to the content of the paper. 

-          This study represents a significant contribution to establishing mutagenesis library ontaining over 4000 M1 seeds of watermelon mutant was constructed by using 60Co γ-rays irradiation, as well characterization seed viability and morphological, physiological and genetic traits of plant M progenies.

-          The main question of paper  addressed to establish  specificity of morphological, physiological and genetic traits of three valuable mutants (M1-3, M2-1, and M1-5), and further their M2 offsprings of plants, providing valuable germplasm resources for further researches on watermelon genetics breeding and functional genes exploring.

-          The aim of research  is clearly and fully pointed  in abstract. Author need clearly pointed out  aim of study  on the end of chapter of Introduction as particular paragraph .

-          Key words are appropriate

-          Scientific methodology is applied correctly for this type of study.

-          Results are clearly presented and discussed.

-          Tables, figures, pictures are clear.

-          The conclusions are clear and based on research results.

-          Manuscript is acceptable after minor corrections!

 Suggestion:  

In line 306  need correct word   chlorophyll-defcient  with chlorophyll-deficient

Author Response

Response to reviewers:

Reviewer #1

This paper correspond for scope of journal. 

The title corresponds to the content of the paper. 

This study represents a significant contribution to establishing mutagenesis library containing over 4000 M1 seeds of watermelon mutant was constructed by using 60Co γ-rays irradiation, as well characterization seed viability and morphological, physiological and genetic traits of plant M progenies.

The main question of paper  addressed to establish  specificity of morphological, physiological and genetic traits of three valuable mutants (M1-3, M2-1, and M1-5), and further their M2 offsprings of plants, providing valuable germplasm resources for further researches on watermelon genetics breeding and functional genes exploring.

The aim of research is clearly and fully pointed in abstract. Author need clearly pointed out aim of study on the end of chapter of Introduction as particular paragraph.

      Many thanks for your kind suggestion.

      Following your advice, we have re-written the sections of Abstract and Introduction in the revised version of this MS.

Key words are appropriate

Scientific methodology is applied correctly for this type of study.

Results are clearly presented and discussed.

Tables, figures, pictures are clear.

The conclusions are clear and based on research results.

Manuscript is acceptable after minor corrections!

      Thank you very much for your positive affirmations of our MS.

Suggestion:  

In line 306  need correct word   chlorophyll-defcient  with chlorophyll-deficient.

      Many thanks for pointing out the typos within our original submission.

      We have corrected this mistake in the new version of this manuscript.

      Thanks again for all your suggestions.

Reviewer 2 Report

Dear authors

The following issues are present:

Abstract

ü  In general, this section is poorly written. It is written simply. This section should include the most important findings from this study. As a result, this section should be improved.

ü  This manuscript contains no novelties.

ü  Before describing the goal, the authors must define the issue in a single line and explain why they chose this approach to study this research.

ü  The aim of study is not correctly mentioned

ü  No information about the type of experimental design and its component is available in this manuscript.

ü  Some quantitative data about the chlorophyll content should be added

ü  In the final line of the abstract, the authors should present a decisive conclusion derived from the research and provide a single line of future prospects.

Keywords

ü  The content of keywords did not reflect the content of this manuscript and the words used for forming the title should not be used as the keywords. So, the structure of keywords should be changed.

Introduction

ü  No information about the method of the chemical composition of watermelon is available

ü  The authors should give some lines about the knowledge gap which their research has covered along with the hypothesis statement

ü  Also, the authors should provide a novelty statement at the end. What new things authors have done or correlated in this research compared to old ones?

ü  The general and specific aim should be specified

Materials and Methods

ü  The authors should write the procedure of irradiation of seeds in detail

ü  The authors should detail the procedure of the measurement of morphological traits

ü  All abbreviations should be written in full name

ü  The equations of chlorophyll a and b should be added

ü  No information about the type of experimental design for phenotypical analysis and its component is available in this manuscript.

Results and discussion

ü  The status of significant of each studied parameter should be mentioned at the beginning of the text

ü  All captions should be improved, showing the contents of tables and figures

ü  The discussion is weak. The authors should interpret all results obtained in this study by adding some information about the results obtained in their study. The authors should explain how all of the findings from this study relate to their own findings. The authors should explain the impact of irradiation on the morphological and physiological traits.

Conclusion

ü  The authors should summarize the most significant findings because they have written this section in an easy-to-read manner.

ü  Future works about this research should also include additional works

moderate correction is needed

Author Response

Response to reviewers:

Reviewer #2

Dear authors

The following issues are present:

       We appreciate for all your valuable suggestions to improve the quality of this MS.

Abstract

  1. In general, this section is poorly written. It is written simply. This section should include the most important findings from this study. As a result, this section should be improved.
  2. This manuscript contains no novelties.
  3. Before describing the goal, the authors must define the issue in a single line and explain why they chose this approach to study this research.
  4. The aim of study is not correctly mentioned.
  5. No information about the type of experimental design and its component is available in this manuscript.
  6. Some quantitative data about the chlorophyll content should be added.
  7. In the final line of the abstract, the authors should present a decisive conclusion derived from the research and provide a single line of future prospects.

         We are sorry for the poor description of this part.

         Following your advice, we have rewritten the Abstract in the revised manuscript.

         To date, research on the effect of 60Co γ-rays irrigation in watermelon is limited. Recently, the optimal dose of 60Co γ-rays had been determined for watermelon library construction [References 23,24]. However, to our knowledge, a large-scale mutant library has not been implemented generated using the 60Co γ-rays irrigation method to date. Therefore, we established a large irrigation library containing 4,000 M1 seeds here. The mutated in functional genes obtained through irradiation randomly occurred. For example, the phenotype of mutant M1-3 with fusiform fruit is different from the elongated fruit shape previously described [Reference 3], indicating its novelty and potential for exploration of new regulatory gene. Meanwhile, the mutants C1-No.3 and C1-No.4 with thickening and thinning pericarp are also new valuable germplasm resources for developing of new watermelon varieties with favorable characteristics and exploring functional genes. In summary, there are at least two innovations in our research. First, we constructed a large-scale mutant library through 60Co γ-rays irrigation method; Second, several mutants with stably inherited morphological phenotypes had not been documented yet in other mutant libraries, which provide new resources for watermelon variety development and gene exploration.

Keywords

  1. The content of keywords did not reflect the content of this manuscript and the words used for forming the title should not be used as the keywords. So, the structure of keywords should be changed.

         Yes. The keywords have been changed according to your suggestion.

Introduction

  1. No information about the method of the chemical composition of watermelon is available.
  2. The authors should give some lines about the knowledge gap which their research has covered along with the hypothesis statement.
  3. Also, the authors should provide a novelty statement at the end. What new things authors have done or correlated in this research compared to old ones?
  4. The general and specific aim should be specified.

        Many thanks for your valuable suggestions to improve the quality of this MS.

        Following your advice, we have made modifications to the description of this Section, especially in the last paragraph.

Materials and Methods

  1. The authors should write the procedure of irradiation of seeds in detail.
  2. The authors should detail the procedure of the measurement of morphological traits.
  3. All abbreviations should be written in full name.
  4. The equations of chlorophyll a and b should be added.
  5. No information about the type of experimental design for phenotypical analysis and its component is available in this manuscript.

         Following your suggestions, the procedure of irradiation of seeds, observation of morphological traits, equations of chlorophyll a and b had been added in the revised version of this MS.

        For all the abbreviations, we have checked and given the full names in the new version of this MS.

        Moreover, this manuscript has been modified according to your advice.

Results and discussion

  1. The status of significant of each studied parameter should be mentioned at the beginning of the text.
  2. All captions should be improved, showing the contents of tables and figures.
  3. The discussion is weak. The authors should interpret all results obtained in this study by adding some information about the results obtained in their study. The authors should explain how all of the findings from this study relate to their own findings. The authors should explain the impact of irradiation on the morphological and physiological traits.

         Many thanks for your comments.

         As for Question #18, we do not have a good understanding. We try to make some modifications. For example, in section of Results ‘3.3 Morphological screening of the mutations identified by 60Co γ-ray irradiation and EMS treatments’, two sentences have been added at the beginning of the first paragraph, to state the significance of this part. If we misunderstand your question, please let us know, and we will modified this MS according to your advice.

        All the captions have been improved in the new version of the MS.

        Following your advice, we have carefully modified the Discussion part. Some comparative sentences between our finding and published results, as well as the novelties of this MS, have been added in the revised version.

Conclusion

  1. The authors should summarize the most significant findings because they have written this section in an easy-to-read manner.
  2. Future works about this research should also include additional works.

        Yes. We have re-written this section in the new version of this MS.

Comments on the Quality of English Language

  1. Moderate correction is needed

        Many thanks for your valuable suggestion.

        Following your suggestion, this manuscript has been greatly modified by a professional English editing service.

        We hope our explanations and modifications can satisfy the reviewers, and the English level has met the journal’s standard.

Reviewer 3 Report

In this manuscript, the authors established an EMS-induced mutagenesis library containing over 4000 M1 seeds from watermelon inbred line ‘M08’, which were irradiated by 350 Gy of 60Co γ-rays for 3 hours. This group have developed and published other EMS-induced mutant library, and identified some mutagenesis lines in M1 generation. Using similar approach, in this study, they characterized eight representative mutants. In general, the study was well presented, but I think that the authors could explain better at the end of introduction the work’s originality in relation to other published studies.

Other points:

Line 26. Include a coma “… color, respectively.”

Lines 86-87. Is this formula correct to calculate the germination rate?

In my view, it is “number of germinated seeds x 100 / total number of seeds”. 

Lines 88, 112, 274. Change “And” by “In addition,” or “Furthermore”

Line 105. What’s is this unit “3500r ??? Is this rpm? If you include rpm instead g force, please give details about the specifications centrifuge used in this analysis.

I don’t found the Table 1 in the result section.

Line 134. Is it really 21 mutation types? In table 2, I counted 20. Similarly, I counted 10 mutation types in table 3, instead 11, as stated in line 152.

In Figure 5, I think that authors want to compare Wt versus M1-5 plants, thus I suggest that they could construct a figure showing the data of  Wt; and M1-5 plants together in X axis for each young and mature leaves; instead young and mature leaves together for each Wt or  M1-5 plants.

Line 240… the mutants “were”

Minor editing of English language required

Author Response

Response to reviewers:

Reviewer #3

In this manuscript, the authors established an EMS-induced mutagenesis library containing over 4000 M1 seeds from watermelon inbred line ‘M08’, which were irradiated by 350 Gy of 60Co γ-rays for 3 hours. This group have developed and published other EMS-induced mutant library, and identified some mutagenesis lines in M1 generation. Using similar approach, in this study, they characterized eight representative mutants. In general, the study was well presented, but I think that the authors could explain better at the end of introduction the work’s originality in relation to other published studies.

       We are deeply grateful for your thoughtful advice.

       Following your suggestion, we have made modifications to the description of the Introduction, especially in the last paragraph.

Other points:

  1. Line 26. Include a coma “… color, respectively.”

      Many thanks for your kindly advice.

      We have corrected this mistake in the revised version of this manuscript.

  1. Lines 86-87. Is this formula correct to calculate the germination rate? In my view, it is “number of germinated seeds x 100 / total number of seeds”. 

         We think the formula in our original manuscript is consistent with the one you had mentioned. For instance, there are 50 germinated seeds in a total of 100 seeds. According to the formula in our original manuscript, the germination rate is 50% (50 / 100 * 100% = 50%). And according to your suggestion, the germination rate is also 50% (50 * 100% / 100 = 50%).

  1. Lines 88, 112, 274. Change “And” by “In addition,” or “Furthermore”.

      Many thanks for your valuable suggestions.

      We have corrected these mistakes following your advice.

      This manuscript has been greatly modified by a professional English editing service following your suggestion.

  1. Line 105. What’s is this unit “3500r ??? Is this rpm? If you include rpm instead g force, please give details about the specifications centrifuge used in this analysis.

        Yes. The unit is g. And we have corrected this unit in the revised manuscript.

  1. I don’t found the Table 1 in the result section.

        Sorry for this mistake.

        The Table 1 was indeed included in the original manuscript when we submitted it, but it was lost during the conversion to journal format.

        We have added it in the revised version of this manuscript.

  1. Line 134. Is it really 21 mutation types? In table 2, I counted 20. Similarly, I counted 10 mutation types in table 3, instead 11, as stated in line 152.

        Thanks for pointing out the typos within our original submission.

         In fact, there are 20 and 10 different types of mutations in M1 and M2 generations, respectively. We had corrected these mistakes in the revised manuscript.

  1. In Figure 5, I think that authors want to compare Wt versus M1-5 plants, thus I suggest that they could construct a figure showing the data of  Wt; and M1-5 plants together in X axis for each young and mature leaves; instead young and mature leaves together for each Wt or  M1-5 plants.

         Many thanks for your valuable advice.

         Following your suggestion, we have re-constructed the Figure 5 and added it in the revised version of this manuscript.

  1. Line 240… the mutants “were”

        Yes. We have corrected this mistake in the revised manuscript.

Comments on the Quality of English Language

  1. Minor editing of English language

      Many thanks for your valuable suggestion.

      Following your suggestion, this manuscript has been greatly modified by a professional English editing service.

      We hope our explanations and modifications can satisfy the reviewers, and the English level has met the journal’s standard.

Reviewer 4 Report

Abstract - The objective of the work is not clear
Line 152 - The authors refer to 11 types of mutations but only mention 10.
Line 228 - Change the order in the caption "Ci" must come before "Tr" to follow the order in which the figures are presented.
Lines 242-244 - the reference to mutation in pollen should be removed.
Lines 231-262 - Very long paragraph, citing several works that do not add important information to the study in question. About half of the references cited in this paragraph could be removed, without prejudice to the discussion. It should be more objective.
Lines 264-296 – Very long paragraph. Consider dividing into 3 (on lines 272 and 287).
Conclusion should not be a repetition of the material, methods and results but rather respond to the objectives of the study. It needs to be rewritten.

Author Response

Response to reviewers:

Reviewer #4

  1. Abstract - The objective of the work is not clear.

      We are sorry for the poor description of this part.

      Following your advice, we have rewritten the Abstract in the revised manuscript.
2. Line 152 - The authors refer to 11 types of mutations but only mention 10.

      Yes. We have corrected this mistake in the revised manuscript. Similarly, the 20 types of mutations discovered in Table 2 were also corrected.

  1. Line 228 - Change the order in the caption "Ci" must come before "Tr" to follow the order in which the figures are presented.

        Many thanks for your valuable comments.

        We have corrected this caption in the new version of this manuscript.
4. Lines 242-244 - the reference to mutation in pollen should be removed.

        Yes. The original sentence in Lines 242-244 have been removed in the revised version of this MS.
5. Lines 231-262 - Very long paragraph, citing several works that do not add important information to the study in question. About half of the references cited in this paragraph could be removed, without prejudice to the discussion. It should be more objective.

       Thank you very much for this valuable advice.

       Following your suggestion, we have divided this paragraph in two parts. Some unrelated sentences and two references have been removed.

  1. Lines 264-296 – Very long paragraph. Consider dividing into 3 (on lines 272 and 287).

        Many thanks.

        Some unrelated sentences in this paragraph have been removed in the revised version of this MS.
7. Conclusion should not be a repetition of the material, methods and results but rather respond to the objectives of the study. It needs to be rewritten.

        Yes, we agree with your points. We have re-written this section in the new version of this MS.

        Thanks again for all your suggestions.

Round 2

Reviewer 2 Report

The authors have been addressed  all comments

moderate correction is needed